# In Vitro Evaluation and Mitigation of Niclosamide’s Liabilities as a COVID-19 Treatment

**DOI:** 10.3390/vaccines10081284

**Published:** 2022-08-09

**Authors:** Jesse W. Wotring, Sean M. McCarty, Khadija Shafiq, Charles J. Zhang, Theophilus Nguyen, Sophia R. Meyer, Reid Fursmidt, Carmen Mirabelli, Martin C. Clasby, Christiane E. Wobus, Matthew J. O’Meara, Jonathan Z. Sexton

**Affiliations:** 1Department of Medicinal Chemistry, College of Pharmacy, University of Michigan, Ann Arbor, MI 48109, USA; 2Department of Internal Medicine, Gastroenterology and Hepatology, Michigan Medicine at the University of Michigan, Ann Arbor, MI 48109, USA; 3Department of Microbiology and Immunology, University of Michigan, Ann Arbor, MI 48109, USA; 4Department of Computational Medicine and Bioinformatics, University of Michigan, Ann Arbor, MI 48109, USA; 5U-M Center for Drug Repurposing, University of Michigan, Ann Arbor, MI 48109, USA

**Keywords:** niclosamide, SARS-CoV-2, COVID-19, polypharmacology, drug repurposing

## Abstract

Niclosamide, an FDA-approved oral anthelmintic drug, has broad biological activity including anticancer, antibacterial, and antiviral properties. Niclosamide has also been identified as a potent inhibitor of SARS-CoV-2 infection in vitro, generating interest in its use for the treatment or prevention of COVID-19. Unfortunately, there are several potential issues with using niclosamide for COVID-19, including low bioavailability, significant polypharmacology, high cellular toxicity, and unknown efficacy against emerging SARS-CoV-2 variants of concern. In this study, we used high-content imaging-based immunofluorescence assays in two different cell models to assess these limitations and evaluate the potential for using niclosamide as a COVID-19 antiviral. We show that despite promising preliminary reports, the antiviral efficacy of niclosamide overlaps with its cytotoxicity giving it a poor in vitro selectivity index for anti-SARS-CoV-2 inhibition. We also show that niclosamide has significantly variable potency against the different SARS-CoV-2 variants of concern and is most potent against variants with enhanced cell-to-cell spread including the B.1.1.7 (alpha) variant. Finally, we report the activity of 33 niclosamide analogs, several of which have reduced cytotoxicity and increased potency relative to niclosamide. A preliminary structure–activity relationship analysis reveals dependence on a protonophore for antiviral efficacy, which implicates nonspecific endolysosomal neutralization as a dominant mechanism of action. Further single-cell morphological profiling suggests niclosamide also inhibits viral entry and cell-to-cell spread by syncytia. Altogether, our results suggest that niclosamide is not an ideal candidate for the treatment of COVID-19, but that there is potential for developing improved analogs with higher clinical translational potential in the future.

## 1. Introduction

Since it emerged as a novel betacoronavirus in late 2019, severe acute respiratory syndrome coronavirus 2 (SARS-CoV-2) has caused a global pandemic [1]. Coronavirus disease 2019 (COVID-19), the disease caused by SARS-CoV-2 infection, presents as varying symptoms with different degrees of severity ranging from dry cough and difficulty breathing to acute cardiac injury and refractory pulmonary failure [2,3]. As of April 2022, COVID-19 has caused the death of over six million individuals worldwide [4] and this death toll continues to increase as new SARS-CoV-2 variants of concern (VOCs) emerge with enhanced transmissibility and increased adaptive immune escape [5].

The deadly impact of COVID-19 has created a need to identify potential antiviral treatments for SARS-CoV-2 infection. This has culminated in the development and FDA authorization/approval of several vaccines [6], and three small-molecule antiviral medications including remdesivir [7], molnupiravir [8], and paxlovid [9]. Unfortunately, because of limited worldwide vaccine availability [10], the modest clinical efficacy of existing antivirals [8,9,11,12], and the potential resistance of SARS-CoV-2 variants [13,14], additional therapeutics are urgently needed to help stop the spread of the virus.

A promising strategy for identifying new therapies with the potential for rapid deployment is drug repurposing, whereby compounds with already established safety profiles and robust supply chains are used to treat other diseases [15]. Since the start of the pandemic, several large-scale drug repurposing screens have been conducted [16,17,18,19,20] and have identified many different potential candidates for the treatment of COVID-19. One of the repurposed drugs, which had potent anti-SARS-CoV-2 efficacy in vitro, was the oral anthelmintic compound niclosamide [17,20].

Traditionally used to treat tapeworm infection, niclosamide has been often repurposed in treating a wide range of diseases including several cancers, bacterial infections, viral infections, type 2 diabetes, non-alcoholic fatty liver disease, and rheumatoid arthritis [21]. This range of potential uses is due to the significant polypharmacology of niclosamide, which is known to act on many different biological targets and is a modulator of the Wnt/b-catenin, mTOR, and JAK/STAT3 signaling pathways among others [21]. Niclosamide is also a weakly acidic lipophilic protonophore that can disrupt pH gradients by shuttling protons across lipid membranes [22] including mitochondria and lysosomes/endosomes. This physiochemical property is responsible for its activity as a mitochondrial uncoupler [23] and contributes to its broad activity against viruses, many of which rely on endosomal-cytoplasmic pH gradients in their life cycle [24].

Niclosamide was also previously identified as a potential antiviral for the related coronavirus SARS-CoV, where it was shown to inhibit viral replication in vitro with low micromolar potency [25]. The mechanism of action (MOA) for niclosamide against SARS-CoV-2 may be more complex and multimodal than for SARS-CoV. Niclosamide has been shown to inhibit SARS-CoV-2 endocytosis mediated entry [26], block viral replication by promoting cellular autophagy [27], and disrupt spike (S) protein-mediated syncytia formation via inhibition of the host cell calcium-dependent scramblase TMEM16F [28]. Given the polypharmacology of niclosamide, it is likely that there are additional factors that contribute to its overall efficacy and complicate its MOA. The degree to which each of these MOAs plays a role in the antiviral efficacy of niclosamide against SARS-CoV-2 is unclear.

While niclosamide was clinically effective as an anthelmintic drug, it has substantial limitations for use as a COVID-19 antiviral including its low oral bioavailability (<10%) and poor water solubility [29,30]. Oral administration of niclosamide at 5 mg/kg in rats reaches a maximal serum concentration (C_max_) of only 354 ± 152 ng/mL [31]. As a result, the concentration of niclosamide in the lungs would likely be too low to achieve therapeutic effect. Another limitation exists in that polypharmacology is generally associated with increased adverse effects for repurposed drugs [32]. Finally, the toxicity of niclosamide is a major concern as it has been previously repurposed as a broad anti-cancer agent and has shown significant cytotoxic/cytostatic effects in vitro [33,34]. A better understanding of the anti-SARS-CoV-2 mechanism of action for niclosamide, including potential toxicity and activity against emerging variants of concern (VOC), is needed to effectively evaluate its clinical potential.

The goal of this study was to expand upon the understanding of niclosamide anti-SARS-CoV-2 activity and its potential as a clinical therapeutic using high-content fluorescence imaging and analysis. Herein, we reveal some of the mechanistic and cell morphological characteristics of niclosamide activity, including an analysis of its cellular toxicity after long-term exposure to a therapeutic antiviral dose. Additionally, we investigate niclosamide antiviral activity against SARS-CoV-2 viral VOCs. We hypothesized that because the fusogenicity amongst SARS-CoV-2 variants is known to be different [35,36,37], the potency/efficacy of niclosamide may also vary between strains. Here we reveal the potency of niclosamide against several variants including WA1 (wildtype), B.1.1.7 (alpha), B.1.351 (beta), P.1 (gamma), and B.1.617.2 (delta). Lastly, we report the in vitro results of a structure–activity relationship (SAR) campaign of 33 niclosamide analogs against SARS-CoV-2 in two different cell lines (VeroE6 and H1437). We used the results from this SAR campaign to reveal additional mechanistic details for niclosamide, and aid with the identification of structural analogs with reduced cellular toxicity. Altogether, our results suggest that niclosamide itself is not a suitable candidate for the treatment of COVID-19, yet there is potential for developing analogs with improved properties for future clinical use.

## 2. Results

### 2.1. Niclosamide Has a Poor Selectivity Index

One major concern regarding the utility of niclosamide as a COVID-19 antiviral is its cytotoxicity in comparison with its antiviral efficacy. Here, we aimed to determine the selectivity index (SI) of niclosamide in vitro against SARS-CoV-2 in two different cell models for infection, VeroE6 and the human lung adenocarcinoma cell line H1437. To assess compound-related toxicity, we evaluated the effects of niclosamide on cells after 72 h of compound exposure. We designed and optimized two separate high-content fluorescence imaging assays in 384-well plate format using the different cell lines and measured cell viability and viral inhibition concurrently. In both assays, we used the detection of viral nucleocapsid (N) protein as a direct marker for SARS-CoV-2 infection and cell count per well as an indicator of cell viability. As summarized in the Figure 1A workflow, VeroE6 or H1437 cells were preincubated with a 10-point 2-fold dilution series of niclosamide (N = 10 replicates per condition) for 24 h and then infected with the SARS-CoV-2 B.1.1.7 variant for an additional 48 h post-infection (p.i.). Following infection, cells were fixed, permeabilized, and stained to identify nuclei and viral N protein. Assay plates were imaged at 10× magnification using a CX5 high content imaging platform (*n*= 9 fields captured per well, 0.4 µM pixel size) and processed using the image segmentation and analysis software CellProfiler. Data from the CellProfiler output were used to determine percent infection and percent viability. Infection data were normalized to the average well-level % N positive for infected controls (mock) in each cell line (Appendix A). Percent viability was determined by normalizing the average well-level cell counts for the infected control (Appendix A). We found that niclosamide has potent 50% maximal inhibition (IC_50_) values of 564 nM for VeroE6 and 261 nM for H1437. However, niclosamide caused a 50% reduction in cell viability (CC_50_) at concentrations of 1050 nM and 438 nM for VeroE6 and H1437, respectively, resulting in poor selectivity indices in both cell lines (1.86 for VeroE6 and 1.67 for H1437). The concentration–response curves for this experiment are shown in Figure 1B (VeroE6) and Figure 1D (H1437). Representative images for infected control, mock, and 10 μM niclosamide conditions are included in Figure 1C (VeroE6) and Figure 1E (H1437). As illustrated in Appendix A, the average percentage of N protein-positive cells in untreated infected controls after 48 h of infection was significantly higher in VeroE6 (69%) than H1437 (9%) indicating more efficient cell-to-cell spread in the former. In conclusion, niclosamide has a low SI in two cell lines of fibroblast origin, representing a liability for therapeutic use.

### 2.2. Niclosamide Potency Is SARS-CoV-2 Variant Dependent

Niclosamide has a complex polypharmacology profile against host-cell pathways, which may contribute to the antiviral efficacy and/or cytotoxicity of the compound and lead to variable responses across SARS-CoV-2 VOCs that rely differentially on these pathways. To evaluate the antiviral efficacy of niclosamide against VOCs, we used a modified infection assay in VeroE6. Exposure to niclosamide was reduced to a 1 h preincubation and the assay window was shortened to 24 h post-infection (Figure 2A) to limit compound toxicity. VeroE6 cells were used as they demonstrated a higher N-protein positivity rate than H1437 cells. We evaluated the antiviral activity of niclosamide against the WA1 (wildtype), B.1.1.7 (alpha), B.1.351 (beta), P.1 (gamma), and B.1.617.2 (delta) variants in 10-point, 2-fold dilution. Ten-point dose–response efficacy experiments showed niclosamide had statistically significant differences in efficacy against VOCs, was most potent against the B.1.1.7 strain (IC_50_ = 298 nM), and least potent against the WA1 strain (IC_50_ = 1664 nM). The full efficacy data for all variants are shown in Figure 2B,C. The CC_50_ for niclosamide in this shortened assay was >10 μM (not shown). These data demonstrate variant-dependent antiviral efficacy of niclosamide.

### 2.3. High-Content Analysis Suggests Inhibition of Entry and Syncytia Formation

Cell morphologic analysis of cells infected with VOCs, under the treatment of niclosamide, revealed several defining characteristics of infection influenced by compound treatment. We quantified these observations using morphological cell profiling analysis. We used data (B.1.1.7 variant in VeroE6) from the viral control (mock) and three different efficacious concentrations of niclosamide around its IC_50_ (156 nM, 313 nM, 625 nM) to reanalyze using a more extensive analysis pipeline that included intensity and area/shape measurements for both nuclear and viral channels. For this analysis, syncytia/individually infected cells were defined as “viral objects.” We determined that treatment with niclosamide decreased the maximum size of syncytia (Figure 3A) consistent with an inhibition of cell-to-cell spread. We also found that treatment reduced the number of individually infected cells within a well (Figure 3B) consistent with an inhibition of viral entry. Finally, we found that the N protein intensity of remaining viral objects increased with escalating concentrations of niclosamide (Figure 3C). The combination of these observations suggests multiple MOA including inhibition of viral entry and cell-to-cell spread resulting in fewer infected cells with dramatically increased cellular viral N protein content. These results provide support for the polypharmacology of niclosamide that contributes to multimodal efficacy against SARS-CoV-2. In addition, these results suggest that niclosamide may not directly inhibit viral replication in vitro.

### 2.4. Structure-Activity Relationship of Niclosamide Analogs versus SARS-CoV-2 Infection

Efficacy and cytotoxicity of 33 previously designed analogs [38] of niclosamide were used to establish a preliminary structure–activity relationship (SAR) profile for niclosamide for anti-SARS-CoV-2 activity in VeroE6 and H1437 cell lines. These analogs were also salicylanilides and had substituent modifications on the nitroaniline and/or chlorosalicyl rings (Figure 4A) intending to improve the selectivity while maintaining antiviral efficacy. Analog structures are shown in Appendix A. Analogs were evaluated using the assay described in Figure 1A. All analogs were tested in 10-point 2-fold dilution series (*n* = 3) from a starting concentration of 20 μM. VeroE6 analog screening was performed using the B.1.1.7 variant at an MOI of 0.1, while H1437 screening was performed using the WA1 variant at an MOI of 1. The results from compound testing are summarized in Table 1, which includes IC_50_ and CC_50_ values for both cell lines. As shown in Table 1, we found that 14 analogs retained IC_50_ values in the nanomolar or micromolar range in VeroE6, while seven (compounds **2**, **3**, **4**, **11**, **12**, **24**, **34**) were efficacious in both VeroE6 and H1437. Four compounds (**2**, **7**, **11**, **24**) showed improved potency and reduced cytotoxicity against VeroE6 compared to niclosamide (Figure 4B–D). Overall, improvements in cytotoxicity were less pronounced against H1437. Notably, the variant dependent potency difference was conserved in H1437 cells and was significantly less potent against WA1 (IC_50_ = 16,770 nM) than B.1.1.7 (IC_50_ = 261 nM), consistent with results in VeroE6 cells. 

Apart from compound **8**, we found that the replacement of the nitro group on the nitroaniline ring was well tolerated (compounds **5**, **6**, **7**, **10**, **24**, and **34**) and improved the selectivity index. We also noted that modification to the chloro position on the salicylic acid ring (R2) was well tolerated and all compounds with only this modification retained antiviral efficacy (compounds **2**, **3**, **4**, **11**, and **12**). Many analogs (compounds **7**, **8**, **10**, **14**–**19**, **22**, **25**, **28**–**30**, and **33**) were found to exacerbate infection in H1437 cells (Appendix A) and showed inverted concentration–response curves at high concentrations. Remarkably, the removal of the hydroxyl group (R1) on the salicylic acid ring (compound **9**) resulted in a complete loss of activity in both VeroE6 and H1437 (Figure 5A). This hydroxyl group has been previously reported as the protonophore responsible for the mitochondrial uncoupling activity of niclosamide [23,38].

Given the drastic loss of activity, we evaluated the anti-SARS-CoV-2 efficacy of other protonophore mitochondrial uncouplers including FCCP [39], 2,4 DNP [40], oxyclozanide [41], and dicumarol [42]. These compounds were evaluated against WA1 and B.1.1.7 variants in VeroE6 cells using the 24 h infection conditions described in Figure 3A. Both FCCP and oxyclozanide showed efficacy in the micromolar range (Figure 5B–F). The potency of these compounds was higher against B.1.1.7 than WA1; however, the difference was more pronounced for niclosamide. These results indicate that the mechanism of action for niclosamide against SARS-CoV-2 is at least partially due to its physiochemical property as a protonophore, implicating energetic stress response pathways in SARS-CoV-2 infection.

## 3. Discussion

There remains an urgent need for COVID-19 therapeutics, which can be used effectively in conjunction with vaccines to prevent the spread of SARS-CoV-2 and COVID-19. While the current mRNA vaccines induce robust humoral immunity, they do not induce significant mucosal immunity to reduce population spread. Small molecule antivirals could provide a significant benefit when administered with SARS-CoV-2 vaccines to redue R_0_. Antiviral drugs can also be effective in the vaccine hesitant, immunocompromised patients or can be delivered to areas outside the vaccine delivery cold chain and represent an important adjuvant to large-scale vaccination. 

The FDA-approved oral anthelmintic drug niclosamide has antiviral activity against SARS-CoV-2 infection in vitro and in vivo [43], which has generated interest in its application for the treatment of COVID-19 and resulted in the conductance of several human clinical trials. However, given its high cytotoxicity, unknown efficacy against SARS-CoV-2 variants, low systemic bioavailability, and significant polypharmacology, we were hesitant to consider niclosamide as a promising antiviral option. In this study, we used high-content imaging of SARS-CoV-2 infected cells to evaluate some of the limitations of niclosamide as a COVID-19 antiviral. We also extended our studies to structural analogs of niclosamide, intending to reveal a preliminary structure–activity relationship profile that could be used for future compound development.

Niclosamide has potent cytotoxic/cytostatic effects when applied directly to cells in vitro [34], suggesting that it may have high acute toxicity in vivo with increased systemic exposure. Clinically, the cytotoxicity is limited by the poor bioavailability of niclosamide, which has low systemic exposure. To evaluate niclosamide toxicity, we used high-content fluorescence imaging to determine a selectivity index for niclosamide in two different cell models including VeroE6 and the more physiologically relevant human lung adenocarcinoma cell line H1437. We found that niclosamide has a very poor selectivity index in both cell lines (SI < 2) after 72 h of compound exposure, suggesting that it would likely have a small therapeutic window clinically even if the compound exposure was high enough in the lungs for antiviral efficacy. Longer durations of exposure to niclosamide at relevant antiviral concentrations are likely to cause significant side effects, which limits clinical application. Further studies are needed to evaluate the safety of niclosamide at antiviral concentrations in vivo.

Our results are consistent with the results from recent clinical studies of niclosamide. Since it was identified as an anti-SARS-CoV-2 agent in vitro, there have been several clinical studies to evaluate the antiviral efficacy and safety of niclosamide. Notably, a recent phase 2 clinical trial using 2 g of orally administered niclosamide for 7 days revealed no statistically significant effect on the duration of the contagious period of SARS-CoV-2 [44]. While niclosamide was well tolerated in this trial, the low efficacy and low adverse event rate are likely because the systemic exposure is lower than what is required to observe antiviral activity or compound-related toxicity. To address the poor oral bioavailability, several different formulations have been developed for niclosamide to improve its exposure to the necessary site of action [29,45]. This has included a formulation as an inhalable/intranasal powder to increase compound exposure in the lungs. Unfortunately, a recent phase 1 safety trial using 50 mg over 2.5 days of inhalable/intranasal niclosamide revealed moderate lung irritation in 59% of participants, which suggests compound-related toxicity may be playing a significant role at higher local concentrations in the lungs [46]. Although niclosamide is generally well tolerated when used as an anthelmintic drug, this is because it has low bioavailability, poor solubility, and stays within the GI tract with very low systemic exposure.

A further limitation for using niclosamide as a COVID-19 therapeutic is its unknown efficacy against the different emerging SARS-CoV-2 VOCs. We determined the efficacy for niclosamide against the WA1 (wildtype), B.1.1.7 (alpha), B.1.351 (beta), P.1 (gamma), and B.1.617.2 (delta) variants in VeroE6 cells. We found that there were significant differences in potency ranging from 298 nM (beta variant) to 1664 nM (wildtype). Interestingly, the trend in potency correlates with the ACE2 binding affinity for the different variants [47]. Variants, including alpha and beta, also have higher fusogenicity than the wildtype variant and are more likely to undergo cell-to-cell spread by syncytia [35], which may help explain the differences in potency.

These results are in contrast with those reported by Weiss et al., which showed no significant difference in potency amongst variants [48]. However, their study used qRT-PCR of viral RNA to determine IC_50_ values, which is far less sensitive than a high-content imaging approach and does not provide information on the clinically relevant endpoint of cell-to-cell spread inhibition. Given the differences in potency amongst SARS-CoV-2 variants of concern, there arises a concern for the rapid development or selection of resistant strains that do not respond to niclosamide treatment. While the emergence of drug resistance is possible for any mechanism of action inhibiting SARS-CoV-2, the pronounced difference between niclosamide’s efficacy amongst the VOCs makes niclosamide resistance inexorable. Further studies to understand the mechanistic differences underlying variant-dependent responses to drugs such as niclosamide may ultimately inform de novo drug development for COVID-19.

To understand the MOA, we used morphological profiling of B.1.1.7 infected VeroE6 cells to evaluate the effect of niclosamide treatment on SARS-CoV-2 infection. We found that niclosamide inhibits the spread of the virus to adjacent cells in a concentration-dependent fashion as indicated by the reduction in the size of viral syncytia. We also observed that the total number of viral objects (individually infected cells or syncytia) decreased with niclosamide treatment, which is consistent with entry inhibition. For example, if niclosamide were only influencing cell-to-cell spread, the total number of viral objects would remain constant and only the size of the syncytia would be affected. While the complete mechanism of action for niclosamide is complex, our results suggest that both inhibition of cell-to-cell spread and entry inhibition play a role in its activity (Figure 6). The degree to which each of the MOAs contributes to efficacy may be different for SARS-CoV-2 variants, which could help explain the differences in potency.

The polypharmacology of niclosamide is a major issue for its utility as a COVID-19 antiviral. Niclosamide is known to influence many different signal transduction pathways and has been implicated in the treatment of a wide range of diseases including several cancers, bacterial infections, viral infections, type 2 diabetes, non-alcoholic fatty liver disease, rheumatoid arthritis, and others. Unfortunately, the mechanism of action for niclosamide remains elusive for the majority of its biological effects. It is often unclear if there is a direct interaction between niclosamide and a molecular target, or if there is an indirect mechanism of action at play [21]. An underlying mechanism for its broad activity may be its ability to act as a protonophore, which has many different downstream effects in cells including disruption of pH gradients, mitochondrial uncoupling, and transcriptional modulation of various gene targets [21]. This mechanistic ambiguity also translates to its antiviral efficacy. It is likely that the antiviral activities of niclosamide (e.g., inhibition of entry, replication, and syncytia formation) are all downstream consequences of its activity as a nonspecific protonophore since the activity was lost following removal of the hydroxyl group. If this is the case, then it may be challenging to separate the undesired off-target effects from the antiviral effects. While our studies suggest that niclosamide and other mitochondrial uncouplers demonstrate anti-SARS-CoV-2 efficacy, further studies are warranted to determine if these mechanisms of action are unified by protonophore activity.

While niclosamide is not an ideal candidate itself, it may represent a promising chemical tool for the development of more specific SARS-CoV-2 inhibitors. In particular, inhibition of cell-to-cell spread by syncytia is an extremely attractive mechanism for inhibition. Syncytia, which are multinucleated bodies resulting from the fusion of adjacent cells, are a key characteristic of SARS-CoV-2 infection and have been observed in many post-mortem histological samples from fatal COVID-19 cases [28,49]. Syncytia formation facilitates the rapid spread of the viral genome between cells [50], which increases the area of infected tissue and may enhance immune system evasion [51]. An inhibitor of cell-to-cell infection such as niclosamide may be clinically useful for the treatment or prevention of COVID-19, especially when cocktailed with other direct-acting antivirals.

In this study, we also tested the antiviral efficacy of 33 structural analogs of niclosamide to establish a preliminary structure–activity relationship profile which could aid in the development of compounds with antiviral efficacy and less off-target effects. We identified seven compounds (compounds **2**, **3**, **4**, **11**, **12**, **24**, and **34**) that were efficacious in both VeroE6 and H1437 cell models and four of which had improved potency and reduced cytotoxicity in VeroE6 (compounds **2**, **7**, **11** and **24**). Consequently, we believe there is a potential for designing better niclosamide analogs with improved properties. Additionally, our structure–activity analysis revealed some mechanistic features of niclosamide. Most noteworthy, the removal of the protonophore hydroxyl group resulted in a complete loss of activity in both cell models. The efficacy of analogs strongly relied on their weakly acidic and lipophilic nature. Analogs with higher predicted acidity (pKa) due to the presence of carboxylic acid substituents were generally completely inactive. We also determined that other protonophores, including FCCP and oxyclozanide, also had anti-SARS-CoV-2 efficacy, suggesting that the ability to disrupt pH gradients is central to the mechanism of action for niclosamide. Niclosamide has been shown to neutralize endo-lysosomal pH gradients, which is believed to be responsible for its broad-spectrum antiviral activity [24]. Our results indicate that this nonspecific mechanism of action also significantly contributes to the activity of niclosamide against SARS-CoV-2.

Overall, the poor selectivity index, low bioavailability, complex polypharmacology, nonspecific protonophore activity, and variant-dependent potency of niclosamide limit its potential as a COVID-19 therapeutic. However, our studies have shown that changes to the salicyl and aniline rings can modulate selectivity and bioavailability while maintaining its activity. Therefore, niclosamide represents a useful chemical probe that can be leveraged in a large-scale SAR campaign to design better analogs in the future.

## 4. Methods

### 4.1. Compounds

Niclosamide, FCCP, 2,4 DNP, oxyclozanide, and dicumarol were obtained from Sigma-Aldrich (St. Louis, MO, USA) and prepared as 10 mM stock solutions in dimethylsulfoxide (DMSO). The 33 structural analogs of niclosamide were obtained from previous studies [38]. Compounds were solubilized at 10 mM in DMSO and were dispensed onto cells using an HPD300e digital compound dispenser. 

### 4.2. Cells and Virus

VeroE6 cells were maintained in Dulbecco’s Modified Eagle Medium (DMEM), and H1437 cells were maintained in RPMI 1640 base medium. Both cell lines were supplemented with 10% fetal bovine serum (FBS) and 1× penicillin-streptomycin solution (15140122, Gibco) and were grown at 37 °C with 5% CO_2_ following standard cell culture procedures. These cell lines were tested for mycoplasma contamination before use and were negative. The following reagents were deposited by the United States Centers for Disease Control and Prevention and were obtained through BEI Resources, NIAID, NIH (Manassas, VI, USA): SARS-Related Coronavirus 2, Isolate USA-WA1/2020, NR-52281, USA/CA_CDC_5574/2020 (B.1.1.7), NR-54011, USA/MD-HP01542/2021 (Lineage B.1.351), NR-55282, Japan/TY7-503/2021 (Brazil P.1), NR-54982, USA/PHC658/2021 (Lineage B.1.617.2), NR-55611. Viral stocks were grown in VeroE6 and titers were determined by TCID50 using the Reed and Muench method [52]. All the work with live SARS-CoV-2 virus was performed in a biosafety level 3 containment lab (BSL3) with the approval of the University of Michigan’s Department of Environment and Health and Safety and the Institutional Biosafety Committee.

### 4.3. Anti-SARS-CoV-2 High Content Bioassays 

Assays were adapted from previous work and optimized for H1437 and VeroE6 cell lines [17,53]. For 48 h infection experiments, VeroE6 and H1437 cells were seeded onto 384-well plates (6057300, Perkin-Elmer, Waltham, MA, USA) at densities of 3000 and 5000 cells per well, respectively, in 50 µL of media. After 24 h of cell attachment at 37 °C and 5% CO_2_, compounds were dispensed directly to the cell plates using an HPD300e digital compound dispenser. All wells were normalized to a constant DMSO concentration of 0.2%, and plates contained both infected and uninfected control wells. After 24 h of preincubation with compounds, cells were inoculated with the indicated SARS-CoV-2 variant at MOIs of 0.1 for VeroE6 and 1 for H1437. Cells were incubated with virus and compounds for an additional 48 h and then fixed with 4% paraformaldehyde for 30 min at room temperature. Cells were then permeabilized with 0.3% Triton-X100 for 15 min and stained with anti-nucleocapsid protein primary antibody (ABIN6952432, Antibodies Online, Aachen, Nordrhein-Westfalen, Germany) at a dilution of 1:2000 overnight at +4 °C. Following primary antibody staining, cells were stained with a dye cocktail containing 1:1000 secondary antibody Alexa-647 (goat anti-mouse, A21235, Thermo Fisher, Waltham, MA, USA) and 10 µg/mL Hoechst 33342 pentahydrate (bis-benzimide) for nuclear labeling for a total of 30 min at room temperature. Cells were stored in PBS before imaging. For 24 h infection experiments, the methods were comparable except that VeroE6 cells were seeded at 5000 cells per well, compounds were preincubated for 1 h instead of 24, and the infection window was 24 h instead of 48. All other inoculation, fixation, and staining procedures were identical. 

### 4.4. High Content Imaging 

Stained assay plates were imaged using both a Thermo Fisher CX5 with a 10×/0.45NA objective lens and a Yokogawa Cell Voyager 8000 (CV8000) microscope with a 20×/1.0NA water immersion lens. Imaging techniques were followed as described previously for detection of nuclei and SARS-CoV-2 nucleocapsid protein [17,53]. A total of *n* = 9 fields per well were imaged for all assay plates, accounting for roughly 80% of the total well area. 

### 4.5. Image Processing

Images were processed using the image segmentation and analysis software CellProfiler 4.0 [54]. Separate pipelines were developed for H1437 and VeroE6 images. Pipelines were used to identify nuclei (Hoechst 33342) and viral objects including multinucleated syncytia and individually infected cells (Alexa Fluor 647) by adaptive otsu thresholding. Similar to previous work, infected cells were identified using the relateobjects module whereby any nucleus contained within a viral object was defined as infected [54]. For morphological profiling of B.1.1.7 infection vs. niclosamide in VeroE6, additional intensity, textural and spatial features were measured using CellProfiler 4.0 for both the nuclear and viral channels.

### 4.6. Concentration Response Analysis and IC_50_/CC_50_ Determination

Field level data were grouped at the well level using Knime [55] and used to determine normalized percent infection and percent viability scores. Raw percent infection per well was determined by taking the ratio of infected nuclei to total nuclei and multiplying by 100. Normalized percent infection was then generated such that “100% infection” was equivalent to the average raw percent infection of the viral control for each plate. Cell counts for the entire plate were normalized and 100% viability was based on the average cell count of the infected DMSO control wells. Concentration–response curves were plotted in GraphPad Prism 9 (GraphPad Software) and fitted using a semi-log 4-parameter variable slope model. IC_50_ and CC_50_ values were extracted from percent infection curves and percent viability curves, respectively. Selectivity indices were determined by taking a ratio of the CC_50_ and IC_50_. 

### 4.7. High Content Imaging Analysis of B.1.1.7 Infection Versus Niclosamide 

Object-level data for nuclei and viral objects (syncytia and individually infected cells) were used to evaluate morphological and phenotypic features of B.1.1.7 infection versus niclosamide including changes in N protein intensity and area of viral objects. Only images for the infected DMSO vehicle control, as well as 3 different concentrations of niclosamide (156 nM, 313 nM, and 625 nM) were included in this analysis. The max viral object area reported in Figure 3A represents the largest viral object observed for each condition including all fields and replicate wells. The mean N protein intensity for infected cells was computed at the object level and the results from Figure 3C include data for cells in each condition.

### 4.8. Statistical Analysis and Hypothesis Testing

All statistical analyses and hypothesis testing was performed using GraphPad Prism 9 (GraphPad Software, San Diego, CA, USA). Specifics for statistical analyses, including sample sizes and other important data are included within the text of figure legends.

## 5. Conclusions

There is still an urgent need for effective anti-SARS-CoV-2 therapeutics due to waning vaccine efficacy, the emergence of variants of concern, and the limited efficacy of existing antivirals. One potential therapeutic option is niclosamide, an FDA-approved anthelmintic compound that has shown promising anti-SARS-CoV-2 activity in cell-based assays. Unfortunately, there are significant barriers to the clinical utility of niclosamide as a COVID-19 therapeutic. Our work emphasizes these limitations by showing that niclosamide has high cytotoxicity at antiviral concentrations, variable potency against variants of concern, and significant polypharmacology as a result of its activity as a nonspecific protonophore. Some of these clinical limitations can be mitigated, however, through structural modifications to the niclosamide scaffold, which we demonstrate through a preliminary structure–activity relationship analysis. Overall, this work shows that niclosamide is not a suitable candidate for the treatment of COVID-19, but that structural analogs with improved drug properties may have higher clinical-translational potential.

## Figures and Tables

**Figure 1 vaccines-10-01284-f001:**
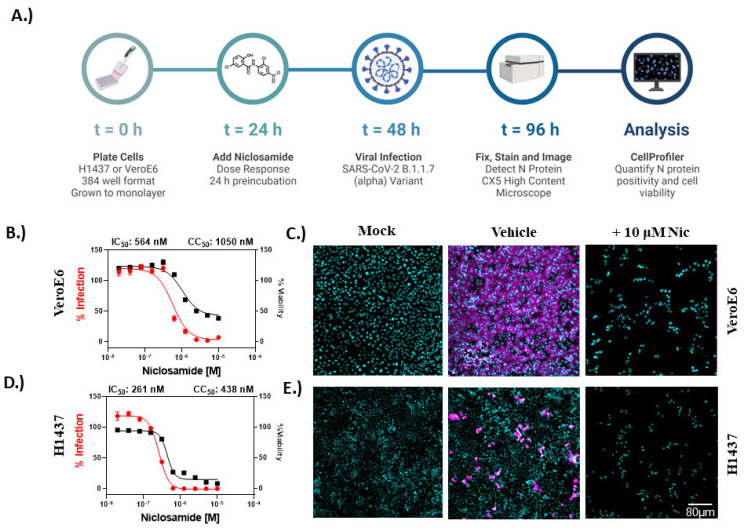
Niclosamide is toxic at antiviral concentrations after long-term exposure. (**A**) Workflow for high content anti-SARS-CoV-2 bioassay screening to determine infection inhibition and cytotoxicity. (**B**) 10-point, 2-fold dilution concentration–response curves for VeroE6 cells infected with B.1.1.7 variant at MOI = 0.1 for 48 h. (**C**) Representative images for mock, vehicle, and 10 µM niclosamide-treated (infected) VeroE6 cells. (**D**) Concentration–response curves for H1437 cells infected with B.1.1.7 variant at MOI = 1 for 48 h. (**E**) Representative images for mock, vehicle, and 10 µM niclosamide-treated (infected) H1437 cells. Data points in concentration–response curves represent mean ± SEM for *n* = 10 replicates per condition. Curve fitting was performed in GraphPad Prism 9 using a semi-log 4-parameter variable slope model. Percent infection is shown using red curves, while percent viability is shown in black. Images were captured at 10× magnification, and the overlays were generated in ImageJ such that cyan = nuclei and magenta = SARS-CoV-2 N protein (uniform scale barb = 80 micrometers).

**Figure 2 vaccines-10-01284-f002:**
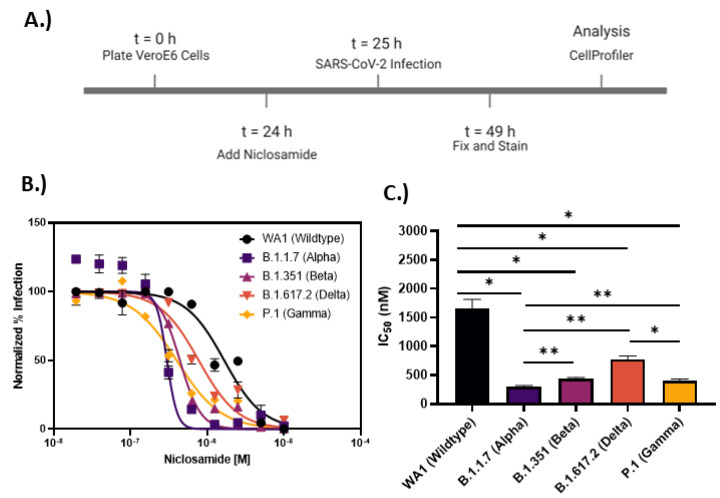
Niclosamide potency is SARS-CoV-2 variant dependent. (**A**) Assay timeline for 24-h infection experiment. The assay window was shortened to reduce niclosamide toxicity. (**B**) 10-point 2-fold concentration–response curves for niclosamide against the different SARS-CoV-2 variants of concern (MOI = 0.1 for each variant) with a top concentration of 10 μM. Curves were fitted with GraphPad Prism 9 software using a semi-log 4-parameter variable slope model. Data for each variant were normalized to the average percent infected of its respective viral control. Data points represent mean ± SEM for *n* = 3 replicates. (**C**) IC_50_ values for niclosamide potency against SARS-CoV-2 variants of concern. Values were extracted from curve fitting using GraphPad 9 and include SEM error bars (WA1: 1664 ± 149 nM, B.1.1.7: 298 ± 23 nM, B.1.351: 440 ± 21 nM, B.1.617.2: 774 ± 58 nM, P.1: 399 ± 34 nM). Significance was determined using Student’s *t*-tests (* = *p* < 0.05, ** = *p* < 0.01).

**Figure 3 vaccines-10-01284-f003:**
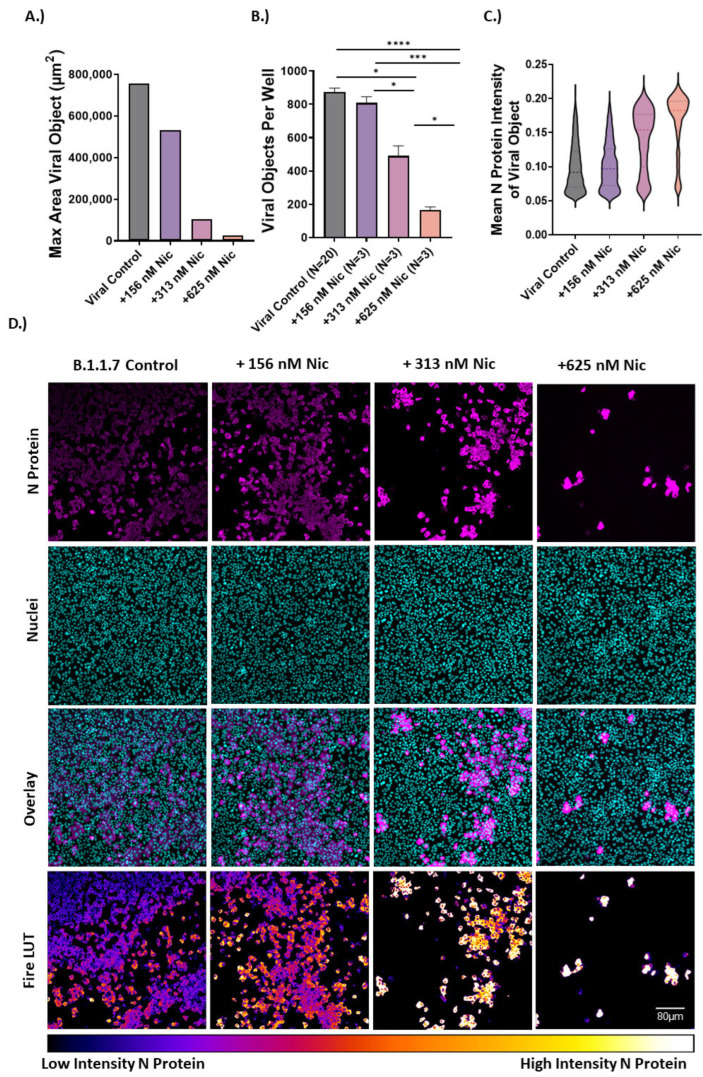
Morphological profiling of B.1.1.7 infection versus niclosamide treatment in VeroE6. Image analysis reveals mechanistic characteristics of niclosamide activity against SARS-CoV-2 infection. (**A**) The maximum area of viral objects decreases with increasing niclosamide concentration. Data are the max area for viral objects in each condition. Viral control: *n* = 17452, +156 nM niclosamide: *n* = 2425, +313 nM niclosamide: *n* = 1470, +625 nM niclosamide: *n* = 496. (**B**) Viral objects per well decrease with increasing niclosamide concentration. Viral objects include single infected cells and syncytia. Replicate values are indicated on the X axis. (**C**) Mean pixel intensity for viral objects in each condition. Pixel intensity increases with increasing niclosamide concentration. (**D**) Representative images for each condition including N-protein channel, nuclear channel, an overlayed image, and a fire lookup table (LUT) image of the N-protein channel. Images were taken on a CX5 high content microscope at 10× magnification. * = *p* < 0.05, *** = *p* < 0.001, **** = *p* < 0.0001. (Scale bar = 80 µM).

**Figure 4 vaccines-10-01284-f004:**
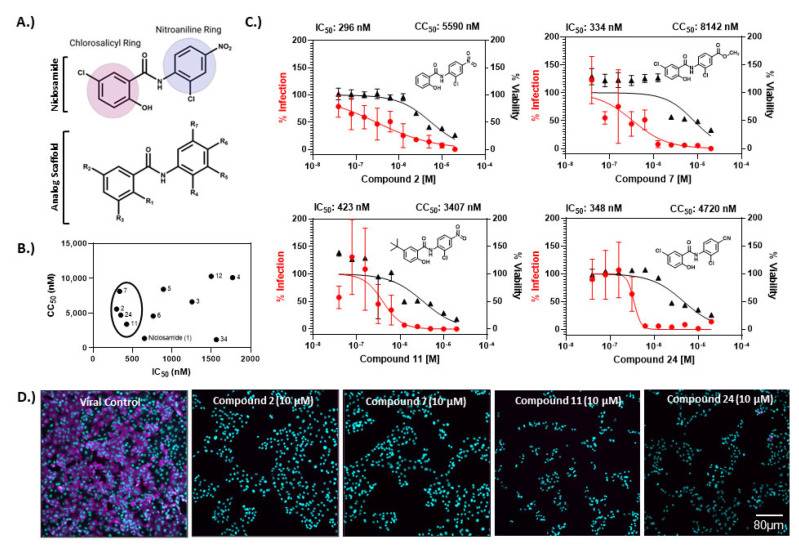
Niclosamide analogs have improved efficacy and reduced cytotoxicity in VeroE6. (**A**) Structure of niclosamide indicating the chlorosalicyl/nitroaniline rings, and analog scaffold with modified substituent positions labeled. (**B**) IC_50_ vs. CC_50_ plot highlighting efficacious compounds in VeroE6. Compounds with improved potency and cytotoxicity profiles are circled on the plot. (**C**) 10-point, 2-fold concentration–response curves for the top four niclosamide analogs with a starting concentration of 20 μM. Data are shown as the mean ± SEM of *n* = 3 replicate wells per condition. Curves for infection (in red) and cell viability (in black) are included. (**D**) Representative images of infected cells treated with indicated compounds and viral control (Vehicle). (10× magnification, cyan = nuclei, magenta = SARS-CoV-2 nucleocapsid protein, scale bar = 80 µm).

**Figure 5 vaccines-10-01284-f005:**
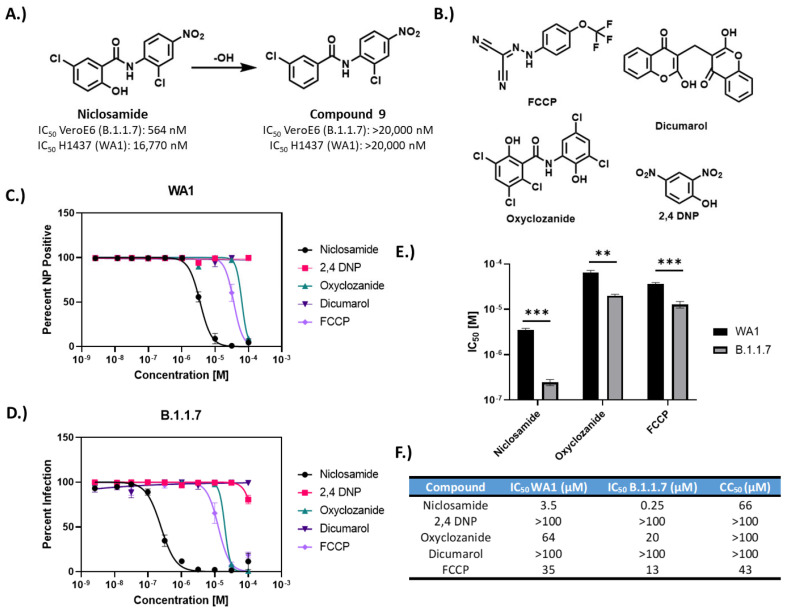
Efficacy of protonophores against SARS-CoV-2. (**A**) Removal of protonophore hydroxyl of niclosamide results in a complete loss of efficacy in both VeroE6 and H1437 cells. (**B**) Chemical structures for other protonophores evaluated against SARS-CoV-2 infection. (**C**,**D**) 10-point, 2-fold concentration–response curves for protonophores versus WA1 variant (**C**) and B.1.1.7 variant (**D**) with starting concentrations of 100 μM. (**E**) IC_50_ plot for antiviral protonophores indicating significantly different potency against variants. ** = *p* < 0.01, *** = *p* < 0.001. (**F**) Table of IC_50_ and CC_50_ values for protonophores.

**Figure 6 vaccines-10-01284-f006:**
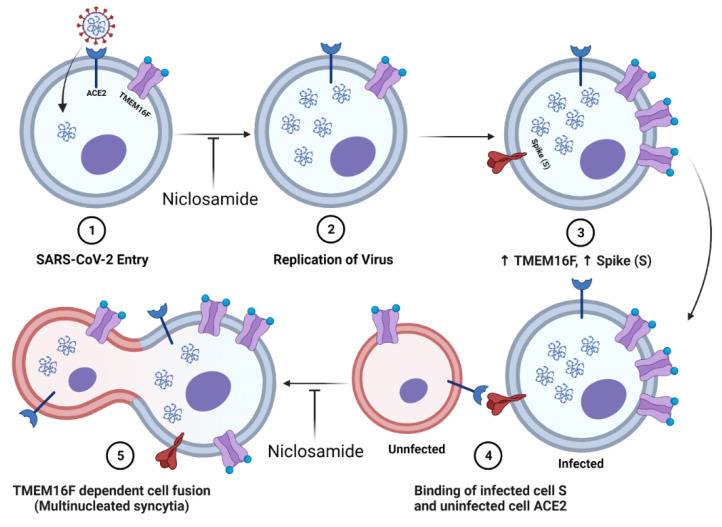
Diagram of niclosamide effect on SARS-CoV-2 entry and spike protein-mediated syncytia formation. (**1**) SARS-CoV-2 binds to the ACE2 receptor of the host cell and enters. Niclosamide has been shown to inhibit this entry step in vitro. (**2**) Viral replication generates many copies of the RNA genome. (**3**) Infection results in an increased expression of viral spike (S) protein and host cell TMEM16F at the plasma membrane. (**4**) The S protein at the surface of an infected cell binds to the ACE2 receptor of an adjacent uninfected cell. (**5**) Spike-dependent syncytia formation is mediated by the calcium-dependent lipid scramblase TMEM16F to generate multinucleated infected cell bodies. Niclosamide, an inhibitor of TMEM16F, has been shown to block spike-dependent syncytia formation.

**Table 1 vaccines-10-01284-t001:** SAR table for niclosamide analogs. IC50 and CC50 values for VeroE6 (SARS-CoV-2 B.1.1.7 variant) and H1437 (SARS-CoV-2 WA1 variant). IC50 and CC50 values were determined using 10-point 2-fold dilution series experiments (*n* = 3) for each compound. Physical properties including molecular weight (MW), cLogP, pKa, and logS were calculated using MOE and included in the table for each compound. Compounds that have efficacy against both cell lines are highlighted in gray.

Compound	R1	R2	R3	R4	R5	R6	R7	Vero-E6 (B.1.1.7)		H1437 (WA1)		MW (g/mol)	cLogP	pKa	logS
								IC_50_ (nM)	CC_50_ (nM)	IC_50_ (nM)	CC_50_ (nM)				
1 (niclosamide)	OH	Cl	H	Cl	H	NO_2_	H	564	1050	16,770	17,060	327.1	4.17	7.98	−5.00
2	OH	H	H	Cl	H	NO_2_	H	296	5590	4915	8403	292.7	3.47	7.52	−4.32
3	OH	CH_3_	H	Cl	H	NO_2_	H	1254	6277	2595	3056	306.7	3.97	7.52	−4.66
4	OH	OCH_3_	H	Cl	H	NO_2_	H	1769	10,110	10,880	8941	322.7	3.39	7.65	−4.39
5	OH	Cl	H	Cl	H	H	H	890	8435	>20,000	18,570	282.1	4.17	8.01	−4.54
6	OH	Cl	H	Cl	H	CH_3_	H	760	4591	>20,000	6048	296.2	4.66	8.01	−4.88
7	OH	Cl	H	Cl	H	COOCH_3_	H	334	8142	Inverted	>20,000	340.2	4.15	7.98	−4.94
8	OH	Cl	H	Cl	H	COOH	H	>20,000	>20,000	Inverted	>20,000	326.1	3.67	4.03	−4.54
9	H	Cl	H	Cl	H	NO_2_	H	>20,000	>20,000	>20,000	>20,000	311.1	4.63	14	−5.27
10	OH	Cl	H	Cl	H	OCH_3_	H	4248	>20,000	Inverted	18,900	312.2	4.05	8.02	−4.59
11	OH	t-Bu	H	Cl	H	NO_2_	H	423	3407	3097	1921	348.8	5.50	7.51	−5.68
12	OH	OCH_3_	H	Cl	H	NO_2_	H	1498	10,290	16,880	7529	322.7	3.39	7.65	−4.39
13	OH	t-Bu	H	Cl	H	COOCH_3_	H	>20,000	14,570	>20,000	>20,000	361.8	5.49	7.50	−5.62
14	OH	t-Bu	H	Cl	H	COOH	H	>20,000	14,250	Inverted	>20,000	347.8	5.00	4.93	−5.22
15	OH	t-Bu	H	F	H	COOH	H	>20,000	>20,000	Inverted	>20,000	331.3	4.41	4.73	−4.69
16	OH	t-Bu	H	CH_3_	H	COOH	H	>20,000	>20,000	Inverted	>20,000	327.4	4.49	4.95	−4.74
17	OH	t-Bu	H	Cl	H	H	COOH	>20,000	>20,000	Inverted	>20,000	347.8	5.00	4.93	−5.22
18	OH	t-Bu	t-Bu	Cl	H	CH3SO_2_N	H	>20,000	>20,000	Inverted	>20,000	453.0	5.98	7.02	−6.39
19	OH	t-Bu	t-Bu	Cl	H	NH_2_	H	>20,000	17,590	Inverted	>20,000	374.9	6.58	7.57	−6.14
20	OH	t-Bu	t-Bu	H	CH_3_	COOH	H	>20,000	>20,000	>20,000	>20,000	383.5	6.52	4.96	−6.10
21	OH	Cy	H	Cl	H	COOCH_3_	H	9910	>20,000	>20,000	>20,000	387.9	5.96	7.50	−6.66
22	OH	Cy	H	Cl	H	COOH	H	4511	>20,000	Inverted	>20,000	373.8	5.48	4.93	−6.26
23	OH	CF_3_	H	CH_3_	H	COOCH_3_	H	>20,000	>20,000	>20,000	>20,000	341.4	4.97	7.49	−5.15
24	OH	Cl	H	Cl	H	CN	H	348	4720	1070	880	307.1	4.27	8.00	−5.03
25	OH	H	t-Bu	Cl	H	COOCH_3_	H	>20,000	>20,000	Inverted	>20,000	361.8	5.22	7.52	−5.52
26	OH	t-Bu	t-Bu	Cl	H	COOH	H	>20,000	>20,000	>20,000	>20,000	403.9	6.77	4.93	−6.48
27	OH	t-Bu	t-Bu	H	CH_3_	COOH	H	16,500	>20,000	>20,000	>20,000	383.5	6.52	4.96	−6.10
28	OH	OCH_3_	H	CH_3_	H	COOH	H	>20,000	>20,000	Inverted	>20,000	301.3	2.38	4.95	−3.46
29	OH	H	t-Bu	Cl	H	COOCH_3_	H	>20,000	>20,000	Inverted	>20,000	361.8	5.22	7.52	−5.52
30	OH	H	t-Bu	Cl	H	COOH	H	>20,000	>20,000	Inverted	>20,000	347.8	4.74	4.93	−5.12
31	OH	H	t-Bu	Cl	H	NO_2_	H	>20,000	7214	>20,000	1110	348.8	5.24	7.52	−5.58
32	OH	H	t-Bu	Cl	H	NH_2_	H	>20,000	>20,000	>20,000	>20,000	318.8	4.55	7.58	−4.78
33	OH	H	t-Bu	Cl	H	NSO_2_CH_3_	H	2879	>20,000	Inverted	>20,000	396.9	3.95	7.03	−5.03
34	OH	Cl	H	Cl	H	Br	H	760	5473	3491	3620	361.0	4.97	8.01	−5.36

## Data Availability

All relevant data are within the paper and its Appendix A.

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
