# Peer review of "In Vitro Evaluation and Mitigation of Niclosamide’s Liabilities as a COVID-19 Treatment"

_vaccines, 2022, doi:10.3390/vaccines10081284_

Round 1
Reviewer 1 Report
The manuscript entitled “in vitro evaluation and mitigation of Niclosamide liabilities as a covid-19 treatment” by wotring et al. really advancement toward the repurposing of the Niclosamide drug for covid-19 treatment. This manuscript has opened a new window to explore Niclosamide. The manuscript is professionally written and conducted detailed studies and discussed the results appropriately. However, the author needs to address the following comments before to accept for publication
Line 27: what is the meaning of B.1.1.7
The author discussed the importance before the introduction. The author needs to discuss it with an editor of the journal about it
lint 54: Abbreviations need to disclose on their first-time appearance example covid-19
Figure 1B: in the left panel of figure 1B, the text font size is not readable. The author needs to increase it
Figure 1B: in the right panel confocal images, the vertical text should start and move to the upper direction. The current text is written from top to bottom. It should be bottom-to-top.
Figure 1B: figure 1B has three figures, the author should label each figure further needed. Just reading the caption of Figure 1B and discussing using figure 1B not clear to the reader which figures discussing exactly.
LINE 151: author mentioned the “concentration response
Figure 3: font size of the text in figures 3A, 3B, 3C axis and legends are completely invisible to read. Need careful revision
Figure 4: the author should remove the box for figure 4C. It is difficult to understand. Better to reorganize accordingly
Table 1: the caption of Table 1 should be at the top of the table.
Author Response
Response to Reviewer 1 Comments:
Line 27: what is the meaning of B.1.1.7
Clarification has been added to the text (alpha variant)
The author discussed the importance before the introduction. The author needs to discuss it with an editor of the journal about it
The importance section has been removed from the start of the manuscript and incorporated as a conclusion at the end of the manuscript
Line 54: Abbreviations need to disclose on their first-time appearance example covid-19
Clarification has been added to the text (Coronavirus Disease 2019)
Figure 1B: in the left panel of figure 1B, the text font size is not readable. The author needs to increase it
Font size has been increased
Figure 1B: in the right panel confocal images, the vertical text should start and move to the upper direction. The current text is written from top to bottom. It should be bottom-to-top.
The orientation of text has been changed to bottom-to-top
Figure 1B: figure 1B has three figures, the author should label each figure further needed. Just reading the caption of Figure 1B and discussing using figure 1B not clear to the reader which figures discussing exactly.
Figures have been relabeled and caption has been modified for clarity
Figure 3: font size of the text in figures 3A, 3B, 3C axis and legends are completely invisible to read. Need careful revision
Font sizes have been increased for better readability
Figure 4: the author should remove the box for figure 4C. It is difficult to understand. Better to reorganize accordingly
The box has been removed from figure 4C
Table 1: the caption of Table 1 should be at the top of the table.
Caption has been moved to the top of the table
Reviewer 2 Report
1. In Supplementary Figure S1B, the group treated with Mock for H1437 cells was measured to have a lower cell survival rate. It would be nice if you could explain about this result.
2. A legend is needed in the left graph in B of Fig 1 of the result. Also, captions such as in red and in black are required.
3. In Figure 1, it is said that niclosamide decreased cell viability (CC50) by 50% in VeroE6 and H1437, and it would be good to attach additional data to see if it reduces cell viability in general cells in addition to the above cells.
4. In Figure 2, Niclosamide was most potent against the B.1.1.7 strain (IC50 = 298 nM), and least potent against the WA1 strain (IC50 = 1664 nM). At this time, what do you think is the reason for the difference in data between WA1 and B.1.1.7(alpha)?
5. In line 242, check the part marked (Figure 4C-F). Figure 4C doesn't have an F, so it's confusing where the result is for that part.
6. Table 1. Comparative experiments were conducted using various compounds of niclosamide. At this table, I wonder additional information on the compounds.
7. Table 1. SAR table for Niclosamide Analogs. I wonder if the data for each compound in Table 1 is a measurement result once. and I think it's right to include Compound 8 in the ‘we found that the replacement of the nitro group on the nitroaniline ring was well tolerated (compounds 5, 6, 7, 10, 24 and 34)’
8. In the data on efficacy and cytotoxicity of Niclosamide analogs in Figure 4, during the experiments on compounds 2, 7, 11, and 24 in (c), the value for %infection showed a large error. I think it's because N=3, but if the N value is higher, the data will be more stable.

Author Response
Response to Reviewer 2 Comments
1.) In Supplementary Figure S1B, the group treated with Mock for H1437 cells was measured to have a lower cell survival rate. It would be nice if you could explain about this result.
The cell count for mock was lower than for infected due to a small plate effect. H1437 cells specifically grow slower on edge wells of 384 well plates despite even seeding density. For the data shown in supplementary figure S1B, the mock conditions were in columns 23 and 24 of the plate, while infected controls were more towards the center of the plate in rows M and N.
2.) A legend is needed in the left graph in B of Fig 1 of the result. Also, captions such as in red and in black are required.
Color designations for the curves have been added to the Figure 1 caption.
3.) In Figure 1, it is said that niclosamide decreased cell viability (CC50) by 50% in VeroE6 and H1437, and it would be good to attach additional data to see if it reduces cell viability in general cells in addition to the above cells.
We agree with the reviewer. While we don’t have CC50 data in other cell lines, cited references 33 and 34 highlights the cytotoxicity of niclosamide in other cell lines.
4.) In Figure 2, Niclosamide was most potent against the B.1.1.7 strain (IC50 = 298 nM), and least potent against the WA1 strain (IC50 = 1664 nM). At this time, what do you think is the reason for the difference in data between WA1 and B.1.1.7(alpha)?
A potential explanation is included in the discussion section of the manuscript. We think that the potency of niclosamide is different for its different mechanisms of action (entry inhibition, syncytia inhibition, etc.). We think niclosamide is most potent against variants which rely heavily on syncytia formation (like B.1.1.7). The process of syncytia formation involves molecular targets such as TMEM16F, which has been shown previously to be inhibited by niclosamide (reference 25)
5.) In line 242, check the part marked (Figure 4C-F). Figure 4C doesn't have an F, so it's confusing where the result is for that part.
The text has been changed to “Figure 4B-D” to highlight figures 4B through 4D
6.) Table 1. Comparative experiments were conducted using various compounds of niclosamide. At this table, I wonder additional information on the compounds.
Molecular weights for each of the compounds have been added to Table 1
7.) Table 1. SAR table for Niclosamide Analogs. I wonder if the data for each compound in Table 1 is a measurement result once. and I think it's right to include Compound 8 in the ‘we found that the replacement of the nitro group on the nitroaniline ring was well tolerated (compounds 5, 6, 7, 10, 24 and 34)
The IC50 and CC50 values were determined using 10-point, 2-fold dilution series experiments for each compound with N=3 replicates per condition. This information has been added to the Table 1. We excluded compound 8 from the list because it did not have efficacy in either cell model. A clarification has been added to the text “Apart from compound 8, we found that the replacement of the nitro group on the nitroaniline ring”
8.) In the data on efficacy and cytotoxicity of Niclosamide analogs in Figure 4, during the experiments on compounds 2, 7, 11, and 24 in (c), the value for %infection showed a large error. I think it's because N=3, but if the N value is higher, the data will be more stable.
We agree with the reviewer that the infection percentage was variable in the lower concentrations of compound treatment. This did not significantly influence the determination of the IC50 values because the error bars narrow dramatically at the onset of efficacy and remain very small to the highest concentrations. We agree that a higher replicate number would produce a more stable infection in the low concentration regime but would not significantly change the IC50 determination.